# Preoperative MRI and Intraoperative Monitoring Differentially Prevent Neurological Sequelae in Idiopathic Scoliosis Surgical Correction, While Curves >70 Degrees Increase the Risk of Neurophysiological Incidences

**DOI:** 10.3390/jcm11092602

**Published:** 2022-05-05

**Authors:** Konstantinos Pazarlis, Håkan Jonsson, Thomas Karlsson, Nikos Schizas

**Affiliations:** 1Stockholm Spine Center, Löwenströmska Hospital, 19489 Upplands Väsby, Sweden; konstantinos.pazarlis@surgsci.uu.se; 2Department of Surgical Sciences, Division of Orthopaedics, Uppsala University, 75185 Uppsala, Sweden; hakan.jonsson@surgsci.uu.se (H.J.); thomas.karlsson@surgsci.uu.se (T.K.); 3Spine Surgery Unit, Department of Orthopaedics, Uppsala University Hospital, 75185 Uppsala, Sweden

**Keywords:** idiopathic scoliosis, MRI screening, intraoperative monitoring

## Abstract

The aim was to investigate the role of preoperative magnetic resonance imaging (MRI) and intraoperative monitoring (IOM) in the prevention of correction-related complications in idiopathic scoliosis (IS). We conducted a retrospective case study of 129 patients with juvenile and adolescent IS. The operations took place between 2005 and 2018 in Uppsala University Hospital. Data from MRI scans and IOM were collected. The patients were divided into groups depending on Lenke’s classification, sex, major curve (MC) size, and onset age. Neurophysiological incidences were reported in ten patients (7.8%), while nine of them had no signs of intraspinal pathology. Six patients (4.7%) had transient incidences; however, in four patients (3.1%), an intervention was required for the normalization of action potentials. Three of them had an MC >70 degrees, which was significantly higher than the expected value. Eight patients (6.1%) had intraspinal pathologies, and two of them (1.5%) underwent decompression. We suggest the continuation of MRI screening preoperatively and, most importantly, the use of IOM. In three cases with no signs of pathology in the MRI, IOM prevented possible neurological injuries. MCs >70 degrees should be considered a risk factor for the occurrence of neurophysiological deficiencies that require action to be normalized.

## 1. Introduction

Idiopathic scoliosis (IS) is a common deformity of the pediatric population with either early or late onset. Cases with onset before the third year of age are defined as infantile idiopathic scoliosis (IIS), cases with onset between 3–9 years old are defined as juvenile idiopathic scoliosis (JIS), and those with onset later than the tenth year of age are defined as adolescent idiopathic scoliosis (AIS) [1].

The prevalence of IS in the pediatric population has been reported in several studies to be between 0.47% and 5.2% [2], while IS represents about 85% of all scoliosis cases [3].

Magnetic Resonance Imaging (MRI) scans are preoperatively used by many physicians to investigate the presence of intraspinal abnormalities that could affect the choice and magnitude of surgery. Such findings are tethered cord, Chiari malformation, and syringomyelia [4]. The presence of the above abnormalities usually lead to a neurosurgical intervention before the correction surgery to prevent spinal cord ischemia or brain stem compression during the correction maneuvers [5,6,7].

However, the use of preoperative MRI in children with IS is debated because the indications vary among physicians. Although MRI is routinely used in many centers, in others it is not. Several studies in the literature strongly recommend the use of preoperative MRI [8,9] whereas others do not support it [10].

Intraoperative monitoring (IOM) with motor-evoked potentials (MEPs) and somatosensory-evoked potentials (SSEPs) is used to detect neurophysiological deficits during corrective surgery for IS [11,12]. The use of IOM helps to preserve the integrity of the spinal cord and prevent neurological sequelae. The surgeon is able to detect neurophysiological abnormalities in real time and act perioperatively. These events can be a misplaced pedicle screw or a very drastic correction. Before the development and use of IOM, the so-called “wake-up test” was used as a method to investigate if patients sustained neurological damage [13,14].

Patients with Cobb’s angle greater that 70 degrees are often discussed as potential candidates for a combined anterior and posterior approach. The reason is limited coronal correction as well as perioperative complications associated with the correction of large curves [15,16,17]. This is probably due to the fact that curves of this size are stiff and the correction effort results in greater perioperative trauma.

This study aimed to investigate the prevalence of intraspinal abnormalities identified by preoperative MRI and how many of those lead to a neurosurgical intervention prior to correction surgery. Furthermore, we registered the incidences of IOM and studied them in relation to the abnormalities in the preoperative MRI.

Moreover, we aimed to investigate potential correlations between findings of IOM and other factors such as the curve size, its location, and its type according to Lenke’s classification [18].

Finally, we studied our patient group for potential differences regarding the IS classification according to Lenke, onset age, sex, curve size, and patient age at the point of surgery.

## 2. Materials and Methods

### 2.1. Study Design

This was a retrospective observational study. The surgical database of our department was used to search for the ICD-10 codes M41.1 and M41.2. These refer to JIS and AIS. The search included patients who underwent surgery for the above diagnoses at the Uppsala University Hospital during the period 2005–2018.

The study was approved by the regional committee of ethics in research (Uppsala County; reference number: 2019-02345). Patients and guardians were informed that the data were to be registered in the Swedish Spine Registry as a routine procedure prior to operation. The study involved retrospective analysis of the data; therefore, the need for informed consent was waived off by the above committee. All of the data management was performed in accordance with the General Data Protection Regulation (GDPR) and Good Clinical Practice (GCP). The patients included in the study received a unique code. and the related data were marked with this code to avoid the possibility of the patients being identified.

The study included patients that matched the above codes for IS with age at onset 3–9 years for JIS and 10–17 years for AIS. The search identified 143 patients, 14 of whom were excluded. The exclusion criteria were inadequate information (care at another hospital, *n* = 4), IIS (*n* = 2), and syndrome/diseases that may have affected scoliosis progression (Marfan *n* = 2, Sotos *n* = 2, Noonan *n* = 1, Prader–Willis *n* = 1, Downs *n* = 1, and Cerebral paresis *n* = 1). The reason for excluding IIS was the limited number of cases (*n* = 2); thus, no conclusions could be drawn based on these.

After the age of onset was determined, the patients were organized into subgroups according to scoliosis onset (JIS or AIS), sex (male or female), age at the time of surgery, and classification according to Lenke [18]. Scoliosis standing plain X-rays were examined, and Cobb’s angle was calculated using the IDS7 version 23 imaging software (Sectra AB, Linköping, Sweden). In order to be able to identify the major curve (MC) and whether a curve was structural or not, we examined X-ray images with side bending. The central sacral vertical line (CSVL) was determined, and the curves were classified according to Lenke. The patients were then divided into three groups. Group 1 included Lenke 1–3 (structural main thoracic curve), Group 2 included Lenke 4 (triple major), and Group 3 consisted of Lenke 5–6 (structural thoracolumbar/lumbar curve). Thus, we generalized Lenke’s classification with these three groups. All of the classifications and measurements of Cobb’s angle and CSVL were reviewed by an experienced spine surgeon. We used the term “extreme curve” for curves ≥70 degrees. The group of patients that showed abnormalities during IOM was organized into two subgroups based on the size of the curve (greater than 70 degrees or not).

All the patients underwent a preoperative screening with neuroaxis MRI (skull base and spinal cord), and the results were collected through the medical records. We only used the MRIs that had been examined by a radiologist. The information was then entered into the study’s database. Data from IOM were reported by a biomedical analyst during each operation and attested by a neurophysiologist.

### 2.2. Statistics

We used the Shapiro–Wilk test to check the population’s normality. The data in the majority of the groups were not normally distributed; therefore, non-parametric tests were used for data analysis. We used Kruskal–Wallis followed by Mann–Whitney U tests. Moreover, Chi-square tables and standardized residuals were used. The results are presented as mean values followed by the standard deviation (SD).

### 2.3. Ethics

The study was approved by the Local Ethics Committee (Dnr 2019-02345) and was conducted in full compliance with the Helsinki Declaration.

## 3. Results

### 3.1. General Description of the Patient Group

The study included 129 patients: 28 males (21.7%) and 101 females (78.3%). Regarding the onset of scoliosis, 58 juveniles (45%) and 70 adolescents (55%) were identified. One patient had unknown onset, and was thus not included in this variable. The mean age at first corrective surgery was 15.1 ± 1.79 years old. The majority had a thoracic MC (90 patients, 70%), and the rest (39 patients, 30%) had a thoracolumbar/lumbar MC. The average size of the MC at operation was calculated according to Cobb’s angle and was found to be 57 ± 12 degrees.

### 3.2. Results of Preoperative MRI Analysis

The preoperative MRI revealed abnormalities in the neuroaxis in eight patients (6.1%). The anomalies identified were tethered cord, Chiari I malformation, lipoma of the filum terminale, and syringomyelia, as shown in Table 1.

### 3.3. Results of IOM

The results are summarized in Table 2. Intraoperative neurophysiological abnormalities were registered in ten patients (7.8%), six of whom were spontaneously normalized. Such transient events might have occurred, for example, if the patient was mispositioned. Five of the patients with transient neurophysiological abnormalities had an MC <70 degrees. The statistical analysis with cross-tables and standardized residuals showed no significant differences between the expected and observed values.

Four patients had neurophysiological abnormalities that required some type of maneuver to be normalized. Only one of these patients had an MC <70 degrees and was the only patient who had an MRI finding of a tethered cord due to previously operated filum terminale lipoma. The other three patients had an MC ≥70 degrees, and this deviated significantly from the expected values (standardized residual = 3.3).

### 3.4. Onset of Scoliosis and Sex Are Related to Age and Major Curve at Surgery

We examined whether there were statistically significant differences regarding the age and size of the curve at surgery using the following three parameters: (a) Lenke’s classification group, (b) onset of scoliosis, and (c) sex. We observed that the age at surgery was associated with sex and with the onset of scoliosis but not with the type according to Lenke. The results are summarized in Table 3.

## 4. Discussion

In this retrospective study, 28 males (21.7%) and 101 females (78.3%) were identified. The difference in sex distribution was quite remarkable, but not completely unexpected. In the literature, the female-to-male IS ratio varies between 1.4:1 and 10:1 (varying with age and curve severity) [1,2]. Notably, male patients were on average one year older than females at surgery, and this age difference was statistically significant. The above observations follow similar patterns to those in the literature where it is described that males tend to be 1–1.4 years older than females at surgery and that the prevalence of IS is greater in females [1,19]. The prevalence of IS in females also increases with age [20]. However, the size of the MC was evenly distributed between male and female patients.

### 4.1. The Diagnosis of Intraspinal Abnormalities Can Lead to Neurosurgical Intervention Prior to Scoliosis Surgery

Among the 129 patients, the preoperative MRI screening revealed intraspinal abnormalities in eight patients (6.2%), two of whom (1.6%) were considered to be in need of a neurosurgical procedure prior to scoliosis surgery (Chiari I + syringomyelia).

There is a large variation in the prevalence (4.2%–14.7%) of spinal anomalies in different studies [9,20,21,22]. Our study shows that the presence of intraspinal anomalies in patients with IS is not uncommon and some of them are candidates for a neurosurgical procedure. Taking this into consideration, this study suggests that patients will benefit from a preoperative MRI screening given the invasiveness and complication risk of surgery.

A type of maneuver was required in four patients for the neurophysiological abnormalities to be normalized. One of these four patients had an MC <70 degrees and was the only patient who had an MRI finding of a tethered cord due to previously operated filum terminale lipoma. The correction in this case was limited to 20 degrees after an intraoperative alteration of MEPs. However, this patient was not eligible for neurosurgical treatment and was the only patient who had both an MRI anomaly and a pathological alteration of the IOM signals.

The rest of the patients with preoperative MRI findings showed no neurophysiological abnormalities intraoperatively. One could argue that the lack of correlation between MRI findings and intraspinal anomalies should not justify the use of preoperative MRI. However, we cannot claim that the presence of intraspinal anomalies in the MRI examination is not associated with intraoperative neurophysiological abnormalities because the radiological anomalies that are considered significant are assessed by neurosurgeons prior to scoliosis surgery.

Intraspinal anomalies can be identified in timepoints where the size of the curve is still small and surgery is not considered as an option. Thus, one could argue about the timing of MRI screening. Early diagnosis and treatment of certain conditions such as Chiari malformation and syringomyelia, when the curves are still small, could prevent progression of the deformity [5,6,7]. For this reason, we have initiated a protocol of MRI screening at the time of initiation of bracing.

### 4.2. Primary Curves Larger than 70 Degrees Could Be a Prognostic Factor for Neurophysiological Deficiencies That Need an Intervention to Be Normalized

Of the four patients with neurophysiological abnormalities in which a maneuver was required, only one had an MC <70 degrees, while the rest had curves ≥70 degrees. This deviated significantly from the expected value according to standardized residuals, implying that the risk of an action-requiring incident to occur is greater in patients with curves ≥70 degrees. To the best of our knowledge, this is the first study to report the above risk after conducting a statistical analysis.

The incidence of neurophysiological events presented in our study is in line with previous studies where neurophysiological incidents were studied in patients that underwent a spinal correction. Nagle et al. showed that 8% of the population had a neurophysiological incident, but only 4% required an intervention [23]. In our study, we registered incidents in 7.8% of the patients, but only 3.1% required an intraoperative action. In line with the results from our study, there is scientific evidence to suggest that the use of intraoperative neurophysiological monitoring is essential to prevent lasting neurological damage as a result of spinal correction [11,24,25]. Therefore, it is still of great clinical value to continue with neurophysiological intraoperative monitoring to avoid neurological damage during scoliosis surgery.

Finally, it is important to point out that the three patients (out of four) who had pathological neurophysiological signals and needed an action to be normalized had no evidence of intraspinal pathology in the MRI examination. These three patients would probably suffer from neurological damage if intraoperative monitoring was not used. Thus, a combination of methods is the most important approach. MRI can lead to preventive neurosurgical measures, and neurophysiological monitoring can prevent neurological damage intraoperatively, especially when MRI screening does not show signs of intraspinal pathology. Therefore, both MRI screening and IOM possess differential roles in spinal cord protection prior to and during IS surgical correction.

### 4.3. Females Have a Higher Prevalence of IS than Males, but Males Tend to Be One Year Older at Surgery

The study of the onset of IS (JIS/AIS) showed statistically significant differences between the groups regarding the age at surgery and the size of the primary curve. Juveniles are operated on average 1.1 years earlier than adolescents. This is likely due to the earlier onset of scoliosis in juvenile patients leading to the early progression and development of larger curves than in adolescents. Before a corrective surgery is decided upon, physicians tend to wait for signs of skeletal maturation to avoid multiple operations and complications related to growing systems [26]. It is reasonable to assume that juveniles reach large values of Cobb’s angle earlier than adolescents, and are thus operated one year earlier on average. Indeed, the size of the MC at surgery is greater in juvenile scoliosis by 4.5 degrees than in adolescent scoliosis. The biggest growth spurt occurs during puberty and varies from person to person. This occurs between 11 and 14 years, with most cases of idiopathic scoliosis progressing and being diagnosed at this period [3]. The age of 11–14 applies primarily to females as the growth spurt for males comes later.

### 4.4. Limitations

We are aware that the patient sample is rather small and the results are not surprising or unexpected. However, we consider it important that the data presented confirm the importance of preoperative MRI and IOM to avoid the risk of perioperative complications.

## 5. Conclusions

The occurrence of intraspinal anomalies that were identified with preoperative MRI in our study is in line with previous studies [9,20,21,22]. Although no association between the detected anomalies and intraoperative neurophysiological abnormalities was found, MRI examination led to preventive neurosurgical action in 1.6% of patients, while IOM protected from neurological sequalae in 2.3%. Therefore, the use of MRI screening prior to surgery in centers where it is not routinely used, as well as the use of IOM during scoliosis surgery, is encouraged. Patients with an MC ≥70 degrees are at greater risk of having a neurophysiological abnormality that requires an intraoperative action to be normalized, and this needs to be taken into consideration especially if the correction is planned to be achieved by a combined anterior and posterior approach [15,16,17].

## Figures and Tables

**Table 1 jcm-11-02602-t001:** MRI findings in conjunction with neurosurgical (NS) treatment prior to scoliosis surgery and IOM deficiencies.

	MRI Finding	NS Treatment	IOM
1	Tethered cord (lipoma)	No (restricted correction)	MEP
2	Chiari I + Syringomyelia	Decompression	No
3	Chiari I + Syringomyelia	No	No
4	Syringomyelia	No	No
5	Syringomyelia	No	No
6	Chiari I	No	No
7	Chiari I	No	No
8	Chiari I + Syringomyelia	Decompression	No

**Table 2 jcm-11-02602-t002:** IOM results.

IOM	MC Size (Degrees)	Number of Patients	Expected Patients	Standardized Residuals
Transient incidences	≥70	1	0.8	0.6
	<70	5	5.2	0.3
Incidences required intervention	≥70	3	0.6	3.3 *
	<70	1	3.2	−1.3

* indicates significance.

**Table 3 jcm-11-02602-t003:** Correlation of scoliosis onset and sex with age and MC at surgery.

		Age at Surgery	*p*-Value	MC Size at Surgery	*p*-Value
Onset of scoliosis	JIS	14.5 ± 2	0.001 *	58.8 ± 11.0	0.004 *
	AIS	15.6 ± 1.4		54.3 ± 10.5	
Gender	Male	15.9 ± 1.3	0.006 *	56.8 ± 13.8	0.842
	Female	14.9 ± 1.8		56.2 ± 10.1	

* indicates significance at *p* = 0.05.

## Data Availability

The clinical database of the study is available from the corresponding author on reasonable request.

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
