# Peer review of "Preoperative MRI and Intraoperative Monitoring Differentially Prevent Neurological Sequelae in Idiopathic Scoliosis Surgical Correction, While Curves >70 Degrees Increase the Risk of Neurophysiological Incidences"

_jcm, 2022, doi:10.3390/jcm11092602_

Round 1

Reviewer 1 Report

Dear authors, unfortunately these are very relevant critical issues.
1) The introduction is too short and does not explain the background
2) In line 71 of materials and methods you state that you have not asked for a ethics committee. Below you have always reported the approval of the ethics committee with reference number. This does not make the study credible
3) The inclusion criteria are not well explained (patients undergoing surgery for LIS or AIS). For example, it would be right to specify an age range
4) Which pathologies that influence the progression of scoliosis are excluded?
5) In line 94 you say that all patients underwent preoperative screening. Why, if the study is retrospective, did these patients have MRI scans? Is it a routine test?
6) In the results of line 9-10, what do you mean by disease onset? How is it possible to diagnose so late? (> 70). You seem to have forced the choice
7) It is not correct statistically correct male and female because they have too different
8) Authors should provide more specific parameters to define a population of patients with scoliosis for whom preoperative MRI is beneficial. It is not a routine test
9) The conclusion is not supported by the results

Author Response

In response to reviewer 1: We are very thankful for the comments given by the reviewer. The manuscript has been revised based on the reviewers' comments. Below you can find a point-by-point answer the reviewer.

  1. Thank you for your comment! The introduction has been revised according to the reviewers´ suggestions. We further added a paragraph about curves larger than 70 degrees (lines 53-57).
  2. Thank you for your comment. We are sorry that the part of ethical approval was not clear in the text and we clarify. The following sentence was added: “The study was approved by the regional committee of ethics in research (Uppsala County; reference number: 2019-02345). Patients and guardians were informed that data will be registered in the Swedish Spine registry as a routine procedure prior to operation. The study involved retrospective analysis of data therefore the need for informed consent was waived off by the above committee
  3. Thank you for the comment on the inclusion criteria. We have added the following sentence in the text: “The study included patients that matched the above codes for IS with age at onset 3-9 years for JIS and 10-17 years for AIS.”
  4. The pathologies that could influence the progression of scoliosis are now included in the text. Please see lines 84-86
  5. Thank you for your comment and thank you for giving us the opportunity to answer on that. It is indeed a retrospective study. We understand that in many scoliosis centres MRI is not used routinely. However, MRI is routinely used preoperatively at Uppsala Akademiska hospital in IS patients.
  6. We use the term “onset of scoliosis” to describe either JIS or AIS. Thank you for pointing out that. We did a change in the text in order to describe the size of the curve at the time of operation (line 125).
  7. We agree with the reviewer and we therefore highlight that the results in males and females slightly differ.

8 and 9. Thank you for your comment. We have revised the discussion and conclusion accordingly. Please see lines (184-189, 248)

Reviewer 2 Report

The aim of the present study was to

evaluate the role of investigate the role of preoperative magnetic resonance imaging  and intraoperative monitoring in the prevention of correction-related complications in idiopathic scoliosis  Although the idea could be interesting, the present investigation   requires substantial improvement.

General

A revision of the English language by a native speaker must be performed. PLEASE DO CORRECT SPELLING MISTAKES.

Introduction

  • Some sentences are confused and need to be rephrased

Material and methods

  • The study design should clearly stated in this section
  • An observational study must follow the STROBE guidelines. A STROBE checklist should be attached to the manuscript as supplemental material (www.strobe-statement.org).
  • How the authors checked the population normality?

Discussion

  • Some sentences are confused and need to be rephrased
  • According the STROBE guidelines the discussion section should give an overview of the literature in the first part and discussing the results that you reached in the second one.

Author Response

We are very thankful for the comments given by the reviewer. The manuscript was meticulously revised according to the reviewers´ suggestions.

The STROBE guidelines were actually used as a checklist for the present study. This is a retrospective observational study. The inclusion criteria, subgroups and statistical methods are described in the materials and methods section.

A limitation section was added in the discussion.

A checklist is attached as supplemental material

Reviewer 3 Report

Presented was a paper on the role of preoperative MRI and intraoperative neuromonitoring in surgical correction of idiopathic scoliosis. Overall, the results are not surprising or to be considered truly new. However, like the authors, I personally consider it important that the data presented confirm the importance of these measures to avoid the risk of perioperative complications. 
The paper is presented in a very well-rounded and readable manner. Scientifically, I do not see any major flaws. The statistical analysis could be more detailed and presented accordingly, but with the number of patients and the expected rarely documented complications, this is not mandatory. 
The only point of criticism I would like to make is the following: 
For a neutral reader it is difficult to understand why the limit of an "extreme" main curvature was chosen at 70 degrees. This is not explained anywhere ( unless I missed it). This should be listed in the methodology and explained in the disscussion, especially since this value is drawn as a key conclusion for a risk constellation. 

Author Response

In response to reviewer 3: We are very thankful for the comments given by the reviewer. The manuscript has been revised based on the reviewers' comments.

The introduction and conclusions have been revised according to the reviewers´ suggestions. We further added a paragraph about curves larger than 70 degrees (lines 53-57, 249-252).

Round 2

Reviewer 1 Report

Dear authors, you have made the required changes and improved the paper.